# Disability among Women and Men Who Married in Childhood: Evidence from Cross-Sectional Nationally Representative Surveys Undertaken in 37 Low- and Middle-Income Countries

**DOI:** 10.3390/ijerph20010088

**Published:** 2022-12-21

**Authors:** Eric Emerson, Gwynnyth Llewellyn

**Affiliations:** 1Centre for Disability Research and Policy, Faculty of Medicine and Health, University of Sydney, Sydney, NSW 2006, Australia; 2Centre of Research Excellence in Disability and Health, University of Sydney, Sydney, NSW 2141, Australia; 3Centre for Disability Research, Faculty of Health & Medicine, Lancaster University, Lancaster LA1 4YW, UK

**Keywords:** children, child marriage, disability, low- and middle-income countries

## Abstract

Child marriage, which the UN’s Sustainable Development Goal seeks to eliminate by 2030, represents a violation of the human rights of children. These concerns are driven by the negative impact of child marriage on the health of children married in childhood and their children. Little is known about the association between child marriage and disability. We sought to estimate the strength of association between disability and child marriage among women and men in middle- and low-income countries (LMICs). Secondary analysis was undertaken of nationally representative samples involving 423,164 women in 37 LMICs and 95,411 men in 28 LMICs. Results were aggregated by random effects meta-analysis and mixed effects multilevel multivariate modelling. The prevalence of disability was significantly greater among women and men who were married in childhood, especially among those married under the age of 16. The strength of these associations varied by age group and age at first marriage. Further research is required to understand the causal pathways responsible for the increased likelihood of disability among women and men married in childhood. National initiatives to eliminate child marriage may need to consider making reasonable accommodations to policies to ensure these are equally effective for women and men with disabilities.

## 1. Introduction

Child marriage (defined by the UN as marriage under the age of 18) is considered by the UN as both a violation of the human rights of children and as a hindrance to national development [1]. Sustainable Development Goal target 5.3 seeks to eliminate the practice of child marriage by 2030. These concerns are driven by the documented impact of child marriage on the health of children married in childhood and their children [2,3,4,5,6,7,8,9,10,11,12], increased exposure to intimate partner violence [2,3], reduced literacy, lower access to education, reduced earning and wealth [2,3,13] and the significant economic costs associated with these consequences [2,3]. However, as recently as 2013 child marriage under 18 was legally permitted in 52% (for girls) and 33% (for boys) of the World’s countries [14].

Between 2010 and 2016, 21% of women aged 20–24 were married (or began a de facto marital relationship) in childhood [15]. Although less attention has been paid to child marriage among boys, it has recently been estimated that 5.4% of men aged 20–24 were married or began a de facto marital relationship when they were still children [16]. While child marriage occurs in high-income countries [17], it is particularly prevalent in LMICs in South Asia and Sub-Saharan Africa [1,15,18,19,20,21], in rural areas [21] and among poorer children and children with lower levels of education [4,18,21,22].

Very little attention has been paid to the relationship between child marriage and disability. UNICEF’s report on progress towards ending child marriage made no mention of disability [1]. UNICEF’s report on the situation of children with disabilities made no mention of child marriage [23]. This is surprising as: (1) child marriage is associated with poorer health and increased risk of exposure to well-established determinants of poorer health and disability [2,3,4,5,6,7,8,9,10,11,12,24]; and (2) the marginalised status of children with disabilities may place them at greater risk of enforced and/or early marriage [25].

To date, we are aware of only one population-based study that has investigated the association between disability and child marriage, which reported that among 20–29 year old women in Ghana child marriage was associated with an increased prevalence of limitations in activities of daily living [24]. Our aim was to estimate the strength of association between disability and child marriage among women and men in a range of middle- and low-income countries.

## 2. Materials and Methods

We undertook secondary analysis of data collected in Round 6 (2017-) of UNICEF’s Multiple Indicator Cluster Surveys (MICS) [26,27]. As an UN programme, all UN member States have the option of participating in MICS. However, the content of MICS is primarily focused on the situation of women and children living in LMICs. Following approval, MICS data were downloaded from http://mics.unicef.org/ (accessed on 8 November 2022). All countries used cluster sampling methods to derive samples representative of the national population of children, women and, in most countries, men. Details of these procedures and arrangements for ethical review used in each country are available at http://mics.unicef.org/ (accessed on 8 November 2022). At the end of the download period (February 2022), nationally representative survey data containing data on adult disability status and age of marriage were available for 37 LMICs (Table 1), representing 27% of all LMICs.

### 2.1. Disabilities

The Washington Group Short Set of Questions on Disability (WGSS) was used to identify disability among participants aged 18–49. The measure is based on self-report of difficulties in six different functional domains (seeing, hearing, walking, remembering/concentrating, self-care, communicating), each with four response options (‘no difficulty’, ‘yes—some difficulty’, ‘yes—a lot of difficulty’, ‘cannot do at all’). WGSS defines disability if the person reports ‘a lot of difficulty’ or ‘cannot do at all’ in one or more domain. Given concern has been expressed about the under identification of people with less severe disability by the WGSS [28,29,30], we also used Bourke’s method to identify respondents with ‘less severe’ disability if they were not identified as having WGSS defined disability but reported ‘some difficulty’ in two or more functional domains. Disability data were missing for 0.01% of respondents.

### 2.2. Child Marriage Status

Informants were asked whether they had ‘ever been married or lived together with someone as if married’ and their age ‘when you started living with your husband/partner’ or ‘first (husband/partner)’. We coded child marriage as yes if they had entered a marital (or de facto marital) relationship under the age of 18 and as no if they had not entered such a relationship. Given several authors have stressed the importance of investigating age at which child marriage occurred [5,31,32], we created two specific groups of child marriage: (1) those married at 16 or 17; (2) those married under the age of 16. Data were missing on marital status for 0.06% of participants. Age at first union information was missing for <0.01% of respondents who had entered such a relationship.

### 2.3. Country Characteristics

Given the association between wellbeing and national wealth in LMICs [33], we used the World Bank 2018 country classification as upper middle income, lower middle income and low income [34]. These classifications are based on per capita Gross National Income adjusted for purchasing power parity (pcGNI; expressed as current USD rates) using the World Bank’s Atlas Method. We downloaded 2018 pcGNI from the World Bank website in May 2020 [35,36].

### 2.4. Household Wealth

MICS data include a within-country wealth index for each household (recoded into within-country quintiles) based on ownership of consumer goods, dwelling characteristics, water and sanitation, and other characteristics that are related to the household’s wealth. The wealth index is assumed to capture underlying long-term wealth through information on the household assets [37,38]. No data were missing for respondents with valid disability data.

### 2.5. Highest Level of Education

The highest level of education received by each participant was recorded using country-specific categories. We recoded these data into a three-category measure: (1) no education; (2) primary education; (3) receipt of secondary or higher-level education. Data were missing for 0.02% of respondents with valid disability data.

### 2.6. Approach to Analysis

First, we estimated the prevalence and predictors of disability and of child marriage. For both we used bivariate descriptive statistics to estimate prevalence (with 95% confidence intervals) in each participating country using the survey data analysis routines in Stata 16 [39] to address the clustered sampling techniques used in MICS and UNICEF’s country-specific person-level inverse probability weights to take account of biases in sampling frames and non-response. We also used mixed effects multilevel multivariate modelling (xtmepoisson in Stata (version 16, StataCorp LLC, College Station, TX, USA) to generate prevalence rate ratios (unbiased estimates of risk) to estimate the association of both disability and child marriage with participant age, highest level of education and within-country household wealth (measured in quintiles) [40].

Second, we estimated the strength of association between disability and child marriage. As above, we report country level data using bivariate descriptive statistics. Given the association between age and the prevalence of disability and the prevalence of child marriage, we used Poisson regression to estimate age-adjusted prevalence rate ratios for the likelihood of child marriage among participants with disability (participants without disability being the reference group). We then provide aggregated results by meta-analysis (using the restricted maximum likelihood (REML) method in Stata 16). Given the high heterogeneity of some of the meta-analyses, as a sensitivity analysis, we aggregated results across countries by mixed effects multilevel multivariate modelling.

Third, to gain a better understanding of the nature of the relationship between disability and child marital status, we stratified the above analyses by participant age group.

All analyses using mixed effects multilevel multivariate modelling specified random effects to allow both the slope and intercept of the relationship between disability and child marriage to vary across countries Given the small amount of missing data, complete case analyses were undertaken. The main analytic sample comprised 423,164 women across 37 LMICs and 95,411 men across 28 LMICs for who valid information on disability and marital/de facto marital status was available.

## 3. Results

### 3.1. Prevalence and Predictors of Disability

Country level estimates of the prevalence of disability are presented in Table 2.

Overall, 14.7% (95%CI 14.4–14.9; inter-country range 4.9–30.2%) of women and 10.5% (95%CI 9.5–11.6; inter-country range 2.6–18.9%) of men were identified as having a disability. Of the respondents with disability, 35.1% (95%CI 34.1–36.2) of women and 37.3% (95%CI 34.8–39.9) of men were identified as having a more severe disability. The risk of disability was significantly greater among participants who were older, poorer and with lower levels of education (Appendix A). Spearman’s non-parametric correlation between country pcGNI and country-level prevalence estimates of disability indicated no significant association between country wealth and the prevalence of disability (women r = −0.10, men r = +0.01).

### 3.2. Prevalence and Predictors of Child Marriage

Information on the prevalence of child marriage for each country is presented in Table 2. Overall, 30.8% (95%CI 29.3–33.2) of women and 7.8% (95%CI 7.2–8.3) of men were identified as being married in childhood, with 15.4% (95%CI 14.8–16.1) of women and 3.3% (95%CI 2.8–3.9) of men being under the age of 16 when married. Likelihood of child marriage was significantly greater among participants who were older, poorer and with lower levels of education (Appendix A). Spearman’s non-parametric correlation between country pcGNI and country-level prevalence estimates of child marriage indicated moderate and statistically significant association between higher country wealth and reduced rates of child marriage (for women r = −0.56, *p* < 0.001 for marriage under 18, r = −0.59, *p* < 0.001 for marriage under 16; for men r = −0.48, *p* < 0.05 for marriage under 18, r = −0.52, *p* < 0.01 for marriage under 16).

### 3.3. Disability and Marriage

Women with disability were 2.5% less likely to have ever entered a marital or de facto marital relationship than women without disability (adjusted prevalence rate ratio (APRR) = 0.975 (95%CI 0.966–0.985), *p* < 0.001). Men with disability were 2.3% less likely to have ever entered a marital or de facto marital relationship than men without disability (APRR = 0.977 (95%CI 0.952–1.00), n.s.).

### 3.4. Disability and Child Marriage

Prevalence of child marriage for women and men with and without disability is presented for each country in Table 3, along with age-adjusted APRRs of the likelihood of participants with disabilities being married in childhood. Marriage under the age of 18 was greater for women with disabilities in 30 of the 37 countries, the difference being statistically significant in 19. Marriage under the age of 16 was greater for women with disabilities in 29 of the 37 countries, the difference being statistically significant in 18. Marriage under the age of 18 was greater for men with disabilities in 16 of the 28 countries, the difference being statistically significant in 7. Marriage under the age of 16 was greater for men with disabilities in 18 of the 28 countries, the difference being statistically significant in 5. In none of the countries with decreased likelihood of child marriage for either women or men was the difference statistically significant.

Aggregation by meta-analysis indicated a significant increased likelihood of marriage under 18 among women with disabilities (APRR = 1.16 [1.10–1.21], *p* < 0.001, I^2^ = 90.4%). Meta-analysis of the association between disability and marriage under 16 was complicated by the failure of within-country likelihood of child marriage estimates for one country (Tuvalu) to resolve due to zero counts in the disability by child marriage 2 × 2 classification. Excluding this country resulted in an estimated APRR of 1.21 (1.14–1.29), *p* < 0.001, I^2^ = 84.2%). As a sensitivity analysis, aggregation by mixed effects multilevel modelling indicated a significantly increased likelihood of marriage under 18 (APRR = 1.22 [1.13–1.31], *p* < 0.001) and under 16 (APRR = 1.26 [1.6–1.37], *p* < 0.001) among women with disabilities. Mixed effects multilevel analysis excluding Tuvalu generated an APRR of 1.26 (1.16–1.37), *p* < 0.001. Additionally controlling for between-group differences in within-country relative household wealth and level of education marginally reduced but did not eliminate the likelihood of child marriage among women with disabilities under 18 (APRR = 1.17 [1.10–1.25], *p* < 0.001), but not under 16 (APRR = 1.20 [1.11–1.30], *p* < 0.001).

Meta-analysis was complicated by the failure of within-country likelihood of child marriage estimates to resolve for marriage under 18 in three countries and marriage under 16 in six countries. Meta-analysis on the subgroups of countries for which estimates were available indicated a significantly increased likelihood of child marriage among men with disabilities (under 18 APRR = 1.31 [1.20–1.42], *p* < 0.001, I^2^ = 2.7%; under 16 APRR = 1.36 [1.21–1.55], *p* < 0.001, I^2^ = 0.0%). Aggregation by mixed effects multilevel modelling, as a sensitivity analysis, generated equivalent APRRs of 1.33 (1.22–1.46), *p* < 0.001 for marriage under 18 and 1.48 (1.31–1.66), *p* < 0.001 for marriage under 16. Additionally controlling for between-group differences in within-country relative household wealth and level of education marginally reduced but did not eliminate the likelihood of child marriage among men with disabilities under 18 (APRR = 1.25 [1.16–1.36], *p* < 0.001), and under 16 (APRR = 1.38 [1.23–1.54], *p* < 0.001).

### 3.5. Analyses Stratified by Age Group

Adjusted prevalence rates estimated by mixed effects multilevel modelling are presented in Figure 1 for women (Figure 2 for men) with disabilities being married under 18 and under 16 by women’s age group.

For both women and men, the relationship between disability and child marriage across age groups varied between those married at age 16–17 and those married below 16. At all eight age groupings, of the women married below age 16, women with disabilities were significantly more likely to be married than their non-disabled peers. Of the women married at age 16 or 17, at all ages except for the youngest age group (18–21) women with disabilities were more likely to be married than their non-disabled peers. These differential patterns were evident for women with more and less severe disability (Appendix A) and for women in upper-middle, lower-middle and low-income countries (Appendix A).

For men married at age 16 or 17, there was no significant relationship between disability status and likelihood of child marriage in the youngest age group (18–21). At older ages (34–41, 46–49), men with disabilities were significantly more likely to be married. Again, in contrast, men with disabilities who married below age 16 were significantly more likely to have been married in the youngest age group (18–21). Additional stratification of these relationships was not undertaken for men due to the markedly smaller sample sizes.

## 4. Discussion

Our analyses of nationally representative samples involving 423,145 women across 37 LMICs and 94,889 men across 28 LMICs indicated that: (1) the prevalence of disability was significantly greater among women and men who were married in childhood, especially among those married under the age of 16; (2) these associations were only marginally attenuated when analyses controlled for the potentially confounding effects of relative household wealth and highest level of education; and (3) among the youngest participants (age 18–21) significantly increased rates of exposure to child marriage among those with a disability was only evident among those married below the age of 16. To our knowledge, this is the first study to examine the association between disability and child marriage among women and men in nationally representative samples.

Given that all data were cross-sectional and age of onset of disability was not recorded, it is impossible to determine causal pathways that underlie this association. As noted, child marriage has been associated with poorer health and increased likelihood of exposure to well-established determinants of poorer health and disability [2,3,4,5,6,7,8,9,10,11,12,24]. Consequently, this association would be expected to lead to an increasing likelihood of disability over time which, discounting any cohort effects, would result in an increasing likelihood of disability with age. Such patterns are evident in Figure 1 and Figure 2, although the increasing likelihood of disability appears to flatten out at older ages. This could reflect the impact of differential mortality given the low life-expectancy for people in LMICs (life expectancy at birth in 1980 for the nine low-income countries participating in the study ranged from 40 years in Sierra Leone to 52 years in Togo).

Limited anecdotal information suggests that children with disabilities may be at increased likelihood of being exposed to child marriage [25]. If true, and again discounting any possible cohort effects, this should be reflected in an increased likelihood of disability among the youngest respondents. This is consistent with the data presented in Figure 1 and Figure 2, but not for girls married at the age of 16 or 17. The differences in likelihood of disability associated with age of marriage may reflect differences in the dynamics of child marriage at different ages with early child marriage possibly being more likely to be an arranged relationship, and later child marriage possibly being more likely to be a consensual relationship among adolescents. In addition to the above processes, unobserved confounders may have independently contributed to increased likelihood of child marriage and increased likelihood of disability. It is worth noting, however, that controlling for two prominent potential confounders (poverty and level of education) had a minimal impact on the strength of association between child marriage and disability.

The primary strengths of the present study lie in the use of well-constructed nationally representative samples from multiple countries with high response rates. The primary limitations are (1) the cross-sectional nature of the data; (2) the failure of MICS to record age of onset of disability; (3) the use of non-random sample of LMICs; (4) the limited number of potential confounders that it was possible to control for in the analyses; and (5) the lack of information on the nature of child marriage (e.g., whether arranged or forced) [41]. In addition, the use of the Washington Group Short Set of Questions on Disability underestimates the prevalence of disability through its failure to identify people whose disability may be associated with mental health related functional limitations [28,30].

## 5. Conclusions

Our analyses indicate that in middle- and low-income countries women and men who were married in childhood were more likely to have a disability than women and men who did not marry in childhood. Further research is required to understand the causal pathways responsible for the increased likelihood of disability among female and male children married in childhood and among their children. However, initiatives to eliminate child marriage should consider making reasonable accommodations to policies to ensure that they are equally effective for girls and boys with disabilities, especially in relation to early arranged, forced or consensual child marriage. In addition, UN monitoring of progress in eliminating child marriage needs to disaggregate data by disability status to help ensure that ‘no one is left behind’ [1] and UN monitoring of the wellbeing of children with disability needs to report on the prevalence of child marriage [23].

## Figures and Tables

**Figure 1 ijerph-20-00088-f001:**
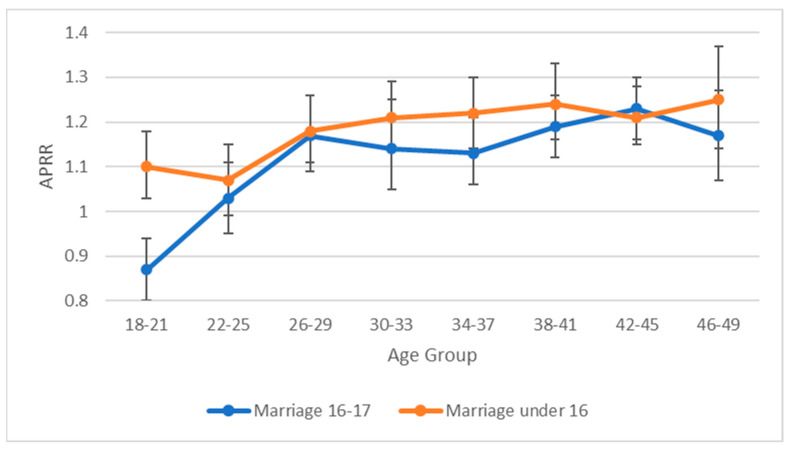
Prevalence rate ratios with 95% CIs (adjusted for within age group variation in age) for women with disabilities having been exposed to child marriage at age 16–17 and under 16 by age group.

**Figure 2 ijerph-20-00088-f002:**
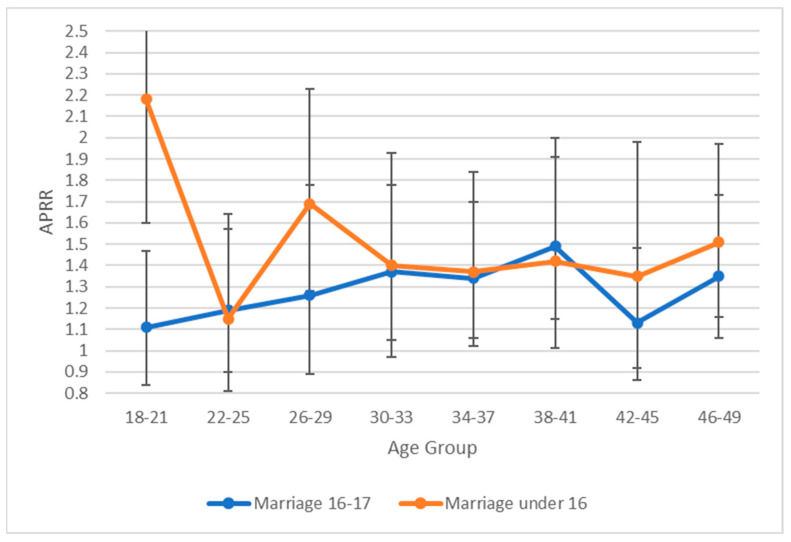
Prevalence rate ratios with 95% CIs (adjusted for within age group variation in age) for men with disabilities having been exposed to child marriage at age 16–17 and under 16 by age group.

**Table 1 ijerph-20-00088-t001:** Country-level survey data for 37 middle- and low-income countries.

Country	Year of Survey	pcGNI (2018)	Response Rate for Women	Sample Size ^a^	Response Rate for Men	Sample Size ^a^
Upper-middle Income						
Costa Rica	2018	USD 11,590	82.5%	6902	n/a	n/a
Montenegro	2018/19	USD 8430	54.9%	2107	39.3%	757
Dominican Republic	2019	USD 7760	97.0%	20,029	n/a	n/a
Cuba	2019	USD 7480	97.7%	8401	95.6%	3456
Turkmenistan	2019	USD 6740	96.0%	6973	n/a	n/a
Guyana	2019/20	USD 6290	84.2%	5290	71.4%	1973
Belarus	2019	USD 5700	93.4%	5270	84.5%	2171
North Macedonia	2018/19	USD 5470	83.8%	2967	n/a	n/a
Tuvalu	2019/20	USD 5430	94.6%	762	94.6%	271
Suriname	2018	USD 5210	74.0%	6261	63.4%	2449
Iraq	2018	USD 5040	98.2%	26,752	n/a	n/a
Georgia	2018	USD 4450	75.4%	6461	57.2%	2476
Kosovo	2019/20	USD 4340	73.6%	4750	59.5%	1850
Tonga	2019	USD 4300	90.3%	2493	83.3%	1048
Lower-middle Income						
Palestine	2019/20	USD 4190	92.9%	9794	n/a	n/a
Samoa	2019/20	USD 4020	93.0%	3659	79.9%	1047
Algeria	2018	USD 3980	91.2%	32,015	n/a	n/a
Mongolia	2018	USD 3660	90.4%	9872	79.8%	4042
Tunisia	2018	USD 3500	93.8%	9788	89.5%	2243
Kiribati	2018/19	USD 3140	96.7%	3806	95.4%	1866
Honduras	2019	USD 2320	86.0%	17,137	77.9%	7933
Ghana	2017/18	USD 2130	97.8%	12,528	96.7%	4309
Sao Tome and Principe	2019	USD 1870	94.8%	2638	87.8%	1160
Zimbabwe	2018/19	USD 1790	92.8%	8888	87.6%	3440
Bangladesh	2019	USD 1750	93.1%	57,699	n/a	n/a
Lesotho	2018	USD 1390	86.0%	5630	80.6%	2425
Kyrgyz Republic	2018	USD 1220	97.2%	5164	n/a	n/a
Nepal	2019	USD 970	98.3%	13,320	97.9%	4856
Low-income						
Guinea-Bissau	2018/19	USD 750	97.6%	9597	92.4%	2391
The Gambia	2018	USD 710	94.0%	11,790	85.3%	3745
Chad	2019	USD 680	98.6%	19,266	97.6%	5674
Togo	2017	USD 660	93.9%	6411	91.4%	1960
Madagascar	2018	USD 510	89.1%	14,872	82.9%	6470
DR Congo	2017/18	USD 490	99.6%	18,978	99.2%	5191
Sierra Leone	2017	USD 490	98.9%	15,649	98.1%	6379
Central African Republic	2018/19	USD 490	92.2%	8122	86.9%	3362
Malawi	2019/20	USD 430	94.6%	21,124	86.5%	5611

Notes: ^a^ Sample sizes are unweighted for women and men aged 18–49 with valid disability information n/a Men’s data not collected pcGNI = World Bank estimate of 2018 per capita Gross National Income (Atlas Method) adjusted for purchasing power parity in US dollars.

**Table 2 ijerph-20-00088-t002:** Country-level estimates of the prevalence of disability and of child marriage in 37 middle- and low-income countries.

	Women			Men		
Country	Prevalence of Disability	Prevalence of Child Marriage (Under Age 18)	Prevalence of Child Marriage (Under Age 16)	Prevalence of Disability	Prevalence of Child Marriage (Under Age 18)	Prevalence of Child Marriage (Under Age 16)
Upper-middle Income						
Costa Rica	25.3% (23.7–27.1)	18.7% (17.1–20.4)	7.5% (6.5–8.6)	n/a	n/a	n/a
Montenegro	6.8% (5.4–8.6)	7.3% (5.6–9.3)	1.9% (1.0–3.6)	6.6% (4.3–10.0)	1.1% (0.4–3.3)	0.2% (0.1–0.9)
Dominican Republic	14.4% (13.5–15.2)	34.7% (33.5–35.9)	18.5% (17.6–19.4)	n/a	n/a	n/a
Cuba	5.9% (4.9–7.0)	28.4% (26.5–30.3)	12.3% (11.1–13.6)	3.9% (3.0–5.2)	8.3% (6.7–9.6)	2.8% (2.2–3.6)
Turkmenistan	4.9% (4.2–5.6)	5.9% (5.2–6.6)	0.8% (0.6–1.0)	n/a	n/a	n/a
Guyana	19.0% (17.3–20.9)	29.3% (27.4–31.3)	12.3% (10.8–13.9)	18.9% (15.9–22.3)	8.9% (7.1–11.0)	3.9% (2.8–5.5)
Belarus	8.2% (7.0–9.4)	5.9% (5/0–7.0)	0.4% (0.2–0.7)	5.5% (4.2–7.2)	1.5% (1.0–2.2)	0.4% (0.2–0.8)
North Macedonia	13.1% (10.6–16.0)	9.7% (7.4–12.6)	3.2% (2.1–4.9)	n/a	n/a	n/a
Tuvalu	14.8% (13.6–16.1)	8.1% (4.9–13.0)	0.7% (0.6–0.7)	15.0% (12.7–17.7)	1.5% (1.2–1.9)	1.5% (1.2–1.9)
Suriname	16.8% (15.8–17.6)	30.2% (29.1–31.3)	12.6% (11.8–13.4)	8.8% (7.7–10.0)	11.2% (10.0–12.5)	3.7% (3.0–4.5)
Iraq	15.6% (14.7–16.6)	24.9% (24.0–25.8)	11.4% (10.6–12/3)	n/a	n/a	n/a
Georgia	21.0% (19.6–22.5)	17.1% (15.7–18.5)	4.3% (3.6–5.0)	12.9% (11.0–15.2)	2.6% (1.9–3.6)	0.5% (0.2–1.0)
Kosovo	17.5% (16.2–18.9)	8.3% (7.2–9.5)	2.4% (1.9–3.2)	9.2% (7.8–10.7)	2.0% (1.3–2.9)	0.5% (0.3–1.0)
Tonga	8.9% (6.8–11.6)	6.4% (5.1–7.9)	1.2% (0.8–1.8)	7.3% (5.1–10.3)	2.3% (1.4–3.7)	0.7% (0.3–1.8)
Lower-middle Income						
Palestine	8.1% (7.3–9.1)	20.6% (19.4–21.8)	5.8% (5.2–6.5)	n/a	n/a	n/a
Samoa	7.4% (6.3–8.8)	8.9% (7.8–10.2)	2.2% (1.7–3.0)	7.4% (5.5–9.8)	2.8% (1.7–4.5)	1.5% (0.9–2.5)
Algeria	15.8% (14.8–16.8)	3.9% (3.6–4.3)	0.8% (0.7–0.9)	n/a	n/a	n/a
Mongolia	22.2% (20.7–23.7)	8.1% (7.3–9.1)	1.1% (0.8–1.4)	17.4% (15.4–19.6)	3.9% (3.1–5.0)	1.4% (0.9–2.2)
Tunisia	25.9% (25.0–26.8)	3.1% (2.8–3.5)	0.5% (0.4–0.7)	11.0% (9.8–12.4)	0.2% (0.1–0.5)	0.1% (0.0–0.3)
Kiribati	20.1% (18.2–22.3)	19.9% (18.3–21.7)	7.7% (6.9–8.8)	17.6% (15.2–20.3)	8.6% (7.4–10.1)	3.0% (2.3–4.0)
Honduras	23.7% (22.8–24.6)	34.1% (33.0–35.2)	16.2% (15.5–17.0)	18.9% (17.9–20.1)	10.9% (10.1–11.7)	3.5% (3.1–4.1)
Ghana	21.8% (20.6–23.1)	24.1% (22.8–25.6)	11.6% (10.7–12.6)	15.5% (13.7–17.5)	5.8% (4.9–6.9)	2.4% (1.8–3.1)
Sao Tome and Principe	24.0% (21.7–26.5)	30.7% (27.8–33.8)	11.8% (10.1–13.6)	10.9% (9.0–13.2)	5.3% (4.0–6.9)	2.7% (1.8–4.0)
Zimbabwe	11.5% (10.8–12.2)	32.2% (30.5–34.0)	11.3% (10.3–12.3)	9.0% (7.9–10.2)	3.6% (2.9–4.4)	1.3% (0.9–1.8)
Bangladesh	10.5% (10.2–10.9)	58.6% (58.0–59.1)	33.7% (33.2–34.2)	n/a	n/a	n/a
Lesotho	12.2% (11.1–13.5)	19.1% (17.7–20.7)	5.0% (4.3–5.8)	10.1% (8.7–11.8)	2.3% (1.7–3.0)	0.7% (0.4–1.2)
Kyrgyz Republic	12.1% (11.2–13.0)	13.1% (12.2–14.0)	1.1% (0.8–1.4)	n/a	n/a	n/a
Nepal	8.1% (7.3–9.0)	37.3% (35.9–38.7)	18.0% (16.9–19.2)	6.1% (4.7–8.0)	13.4% (10.8–16.4)	5.3% (4.0–6.8)
Low-income						
Guinea-Bissau	7.5% (6.3–8.8)	29.0% (26.9–31.1)	15.1% (13.7–16.5)	2.6% (1.8–3.6)	3.8% (2.9–4.9)	2.0% (1.4–2.7)
The Gambia	8.5% (7.7–9.3)	33.1% (31.3–35.0)	17.8% (16.6–19.1)	8.9% (7.6–10.3)	1.4% (1.0–2.0)	0.6% (0.4–1.1)
Chad	16.7% (15.4–18.0)	53.7% (52.5–54.9)	32.9% (31.8–34.0)	7.8% (6.7–9.1)	9.4% (8.3–10.6)	4.4% (3.6–5.2)
Togo	21.4% (19.6–23.3)	24.7% (23.0–26.5)	11.3% (10.1–12.5)	12.2% (10.4–14.4)	5.4% (4.3–6.9)	2.8% (2.0–4.0)
Madagascar	21.5% (20.5–22.6)	37.5% (36.1–39.0)	19.4% (18.4–20.6)	10.7% (9.6–11.9)	11.8% (10.8–12.9)	4.6% (3.9–5.3)
DR Congo	12.5% (11.0–14.1)	30.8% (28.9–32.9)	15.0% (13.8–16.2)	8.1% (6.7–9.9)	6.8% (5.7–8.1)	3.1% (2.4–4.0)
Sierra Leone	5.4% (5.1–5.8)	34.6% (33.9–35.3)	22.3% (21.7–23.0)	3.2% (2.8–3.7)	12.0% (11.2–12.8)	8.5% (7.8–9.2)
Central African Republic	30.2% (28.5–32.0)	56.9% (55.2–58.5)	37.5% (35.8–39.2)	12.1% (10.5–13.9)	18.0% (16.4–19.8)	9.0% (7.9–10.2)
Malawi	11.5% (10.8–12.3)	42.1% (41.0–43.1)	17.8% (17.0–18.6)	13.0% (11.7–14.4)	8.1% (7.2–9.1)	3.5% (2.9–4.1)

Notes: n/a Men’s data not collected.

**Table 3 ijerph-20-00088-t003:** Prevalence and Risk of Child Marriage by Disability Status.

Country	Child Marriage Under 18			Child Marriage Under 16		
	Women with Disability	Women with no Disability	APRR	Women with Disability	Women with no Disability	APRR
Upper-middle Income						
Costa Rica	25.0% (22.0–28.1)	16.6% (14.8–18.5)	1.48 *** (1.25–1.75)	10.4% (8.6–12.6)	6.5% (5.4–7.7)	1.58 ** (1.21–2.06)
Montenegro	15.1% (8.8–24.8)	6.7% (5.0–8.9)	2.00 * (1.09–3.67)	1.5% (0.5–4.7)	1.9% (0.9–3.8)	0.95 (0.27–3.39)
Dominican Republic	36.4% (33.8–39.1)	34.4% (33.2–35.6)	1.04 (0.96–1.13)	19.8% (17.9–22.0)	18.2% (17.3–19.2)	1.07 (0.96–1.20)
Cuba	38.7% (31.1–47.0)	27.7% (25.9–29.6)	1.37 ** (1.11–1.69)	21.5% (15.1–29.7)	11.7% (10.6–13.0)	1.79 ** (1.29–2.48)
Turkmenistan	8.8% (6.0–12.8)	5.7% (5.1–6.5)	1.57 * (1.04–2.37)	1.5% (0.5–3.9)	0.7% (0.6–1.0)	2.18 (0.70–6.80
Guyana	31.9% (27.7–36.4)	28.7% (26.7–30.9)	1.14 (0.98–1.33)	13.9% (10.9–17.5)	11.9% (10.3–13.7)	1.20 (0.92–1.58)
Belarus	6.1% (3.7–10.0)	5.9% (5.0–7.0)	0.87 (0.51–1.49)	0.0% (0.0–0.4)	0.5% (0.3–0.8)	0.20 (0.04–1.02)
North Macedonia	25.5% (18.9–33.4)	7.3% (5.4–9.8)	3.01 *** (2.13–4.26)	8.5% (5.1–13.7)	2.4% (1.5–3.8)	3.29 ** (1.90–5.69)
Tuvalu	8.4% (4.0–16.9)	8.0% (5.0–12.5)	0.80 (0.40–1.59)	0.0% (0.0–11.7)	0.8% (0.7–0.8)	n/a
Suriname	37.5 (34.6–40.5)	30.1% (28.9–31.4)	1.28 *** (1.15–1.43)	16.7% (14.6–19.1)	13.3% (12.4–14.2)	1.33 ** (1.13–1.57)
Iraq	29.7% (24.8–31.6)	24.0% (23.0–25.0)	1.34 *** (1.23–1.45)	15.1% (13.4–17.1)	10.8% (10.0–11.6)	1.53 *** (1.35–1.73)
Georgia	20.1% (17.3–23.3)	16.3% (14.8–17.8)	1.16 (0.97–1.39)	5.3% (3.9–7.2)	4.0% (3.3–4.8)	1.24 (0.87–1.77)
Kosovo	13.8% (11.3–16.9)	7.1% (6.1–8.2)	1.54 *** (1.22–1.94)	5.9% (4.3–8.1)	1.7% (1.2–2.4)	2.84 *** (1.69–4.76)
Tonga	5.5% (2.8–10.6)	6.5% (5.1–8.1)	0.88 (0.42–1.86)	0.4% (0.0–1.9)	1.2% (0.8–1.9)	0.35 (0.07–1.69)
Lower-middle Income						
Palestine	34.2% (29.4–39.5)	19.4% (18.3–20.5)	1.34 *** (1.15–1.58)	11.3% (8.2–15.4)	5.3% (4.8–5.9)	1.42 * (1.02–1.99)
Samoa	9.8% (6.7–14.0)	8.9% (7.8–10.1)	1.07 (0.75–1.53)	1.5% (0.5–4.8)	2.3% (1.7–3.1)	0.65 (0.20–2.15)
Algeria	5.8% (5.0–6.6)	3.6% (3.2–3.9)	1.42 *** (1.22–1.65)	1.4% (1.0–1.8)	0.7% (0.6–0.8)	1.54 * (1.07–2.22)
Mongolia	9.4% (7.7–11.5)	7.8% (6.8–8.9)	1.28 (1.00–1.65)	1.9% (1.2–3.1)	0.8% (0.6–1.1)	2.51 ** (1.39–4.51)
Tunisia	5.0% (4.2–5.9)	2.5% (2.2–2.9)	1.50 ** (1.19–1.90)	0.8% (0.5–1.2)	0.4% (0.3–0.6)	1.33 (0.73–2.42)
Kiribati	24.1% (21.0–27.5)	18.9% (17.1–20.8)	1.22 * (1.04–1.43)	10.4% (8.2–13.0)	7.0% (6.0–8.2)	1.28 (0.98–1.67)
Honduras	35.8% (34.0–37.6)	33.5% (32.3–34.8)	1.06 * (1.00–1.12)	16.8% (15.5–18.2)	16.0% (15.2–16.9)	1.06 (0.96–1.16)
Ghana	28.2% (25.9–30.7)	23.0% (21.6–24.5)	1.15 ** (1.05–1.26)	14.1% (12.3–16.1)	11.0% (10.0–12.0)	1.20 * (1.03–1.39)
Sao Tome and Principe	29.4% (24.7–34.5)	31.1% (28.2–34.2)	0.92 (0.78–1.08)	11.7% (9.0–15.0)	11.8% (10.1–13.8)	0.95 (0.72–1.24)
Zimbabwe	34.2% (30.9–37.7)	32.0% (30.2–33.8)	1.08 (0.98–1.19)	14.0% (11.5–16.9)	10.9% (10.0–11.9)	1.22 * (1.01–1.48)
Bangladesh	70.5% (69.1–71.8)	57.2% (56.6–57.7)	1.13 *** (1.11–1.16)	45.8% (44.4–47.3)	32.3% (31.8–32.8)	1.21 *** (1.17–1.25)
Lesotho	18.3% (15.2–21.9)	19.2% (17.7–20.9)	0.88 (0.72–1.07)	4.4% (3.0–6.3)	5.1% (4.4–6.0)	0.75 (0.50–1.11)
Kyrgyz Republic	16.6% (13.9–19.7)	13.3% (12.3–14.3)	1.14 (0.93–1.41)	1.1% (0.5–2.2)	1.1% (0.8–1.4)	0.96 (0.43–2.16)
Nepal	42.6% (38.9–46.4)	36.8% (35.4–38.3)	1.06 (0.96–1.17)	22.6%/1104	18.7%/12216	1.11 (0.96–1.29)
Low-income						
Guinea-Bissau	35.6% (30.7–40.8)	28.4% (26.4–30.6)	1.23 ** (1.08–1.40)	22.8% (18.7–27.4)	14.5% (13.2–15.8)	1.53 *** (1.28–1.84)
The Gambia	34.4% (30.5–38.5)	33.0% (31.1–34.9)	0.96 (0.85–1.08)	18.8% (15.7–22.2)	17.7% (16.5–19.1)	0.95 (0.80–1.13)
Chad	54.4% (51.8–57.0)	53.5% (52.2–54.8)	1.04 (0.99–1.09)	34.4% (32.1–36.7)	32.6% (31.5–33.8)	1.09 * (1.01–1.16)
Togo	25.8% (22.5–29.5)	24.4% (22.5–26.4)	1.06 (0.91–1.23)	13.5% (11.2–16.1)	10.7% (9.4–12.1)	1.25 * (1.00–1.55)
Madagascar	37.3% (34.9–39.8)	37.5% (36.0–39.1)	1.06 (0.99–1.13)	18.5% (16.7–20.3)	19.7% (18.6–20.9)	1.02 (0.92–1.13)
DR Congo	37.5% (34.1–41.2)	29.9% (27.9–31.9)	1.24 *** (1.13–1.36)	21.5% (18.8–24.4)	14.1% (12.9–15.3)	1.48 *** (1.28–1.70)
Sierra Leone	44.3% (41.0–47.7)	36.5% (35.7–37.3)	1.17 ** (1.05–1.30)	28.8% (25.8–32.0)	23.5% (22.8–24.2)	1.16 * (1.02–1.32)
Central African Republic	58.5% (55.8–61.2)	56.2% (54.3–58.0)	1.07 * (1.01–1.13)	39.6% (36.8–42.4)	36.6% (34.7–38.4)	1.09 * (1.01–1.18)
Malawi	42.3% (39.8–44.9)	42.0% (40.9–43.2)	0.97 (0.91–1.04)	20.6% (18.5–22.7)	17.4% (16.6–18.3)	1.08 (0.97–1.21)
	Men with disability	Men with no disability	APRR	Men with disability	Men with no disability	APRR
Upper-middle Income						
Montenegro	0.0% (0.0–8.6)	1.1% (0.4–3.6)	n/a	0.0% (0.0–8.6)	0.2% (0.0–1.1)	n/a
Cuba	6.4% (2.9–13.7)	8.1% (6.7–9.7)	0.76 (0.34–1.71))	1.5% (0.3–7.4)	2.9% (2.2–3.7)	0.50 (0.09–2.62)
Guyana	10.0% (6.5–15.2)	8.6% (6.8–10.9)	1.26 (0.82–1.96)	4.5% (2.3–8.7)	3.8% (2.7–5.4)	1.20 (0.60–2.40)
Belarus	0.0% (0.0–4.4)	1.5% (1.0–2.3)	n/a	0.0% (0.0–4.4)	0.4% (0.2–0.8)	n/a
Tuvalu	2.0% (0.2–15.7)	1.4% (0.6–3.2)	1.98 (0.20–19.90)	2.0% (0.2–15.7)	1.4% (0.6–3.2)	1.98 (0.20–19.90)
Suriname	16.3% (12.1–21.6)	11.8% (10.5–13.2)	1.61 ** (1.15–2.27)	7.3% (4.6–11.4)	4.3% (3.5–5.2)	2.01 ** (1.20–3.38)
Georgia	5.3% (2.5–11.0)	2.2% (1.6–3.2)	1.81 (0.83–3.94)	1.1% (0.3–4.3)	0.4% (0.2–1.0)	2.73 (0.0.47–15.68)
Kosovo	1.7% (0.5–5.3)	2.0% (1.3–3.1)	0.65 (0.18–2.31)	0.0% (0.0–2.2)	0.6% (0.3–1.1)	n/a
Tonga	2.0% (0.6–6.6)	2.3% (1.3–3.9)	0.78 (0.20–2.92)	0.8% (0.2–3.5)	0.7% (0.3–1.9)	0.97 (0.17–5.50)
Lower-middle Income						52
Samoa	4.0% (1.3–11.8)	2.7% (1.6–4.6)	1.31 (0.37–4.70)	0.0% (0.0–4.9)	1.6% (0.9–2.8)	n/a
Mongolia	3.6% (2.0–6.4)	4.0% (3.0–5.3)	0.94 (0.47–1.88)	1.5% (0.6–3.9)	1.3% (0.8–2.3)	1.38 (0.46–4.17)
Tunisia	0.0% (0.0–1.6)	0.3% (0.1–0.7)	n/a	0.0% (0.0–1.6)	0.2% (0.1–0.5)	n/a
Kiribati	11.6% (8.1–16.2)	8.0% (6.8–9.5)	1.47 * (1.01–2.15)	3.8% (2.2–6.4)	2.9% (2.1–3.9)	1.34 (0.76–2.39)
Honduras	13.8% (11.9–16.0)	10.2% (9.4–11.1)	1.37 *** (1.15–1.63)	4.7% (3.6–6.2)	3.3% (2.8–3.8)	1.47 * (1.07–2.04)
Ghana	8.6% (5.9–12.4)	5.3% (4.4–6.4)	1.52 (1.00–2.33)	3.4% (2.0–5.8)	2.2% (1.7–2.9)	1.49 (0.81–2.74)
Sao Tome and Principe	4.0% (1.8–8.6)	5.4% (4.1–7.1)	0.68 (0.32–1.45)	2.4% (0.9–6.2)	2.8% (1.8–4.2)	0.86 (0.29–2.55)
Zimbabwe	4.5% (2.7–7.5)	3.5% (2.8–4.3)	1.02 (0.56–1.85)	2.5% (1.2–5.1)	1.2% (0.8–1.7)	1.45 (0.60–3.53)
Lesotho	1.3% (0.5–3.2)	2.4% (1.7–3.2)	0.47 (0.17–1.30)	1.0% (0.3–3.0)	0.7% (0.4–1.3)	1.17 (0.0.30–4.27)
Nepal	22.7% (19.6–26.1)	12.8% (12.1–13.5)	1.60 *** (1.34–1.89)	11.1% (8.9–13.8)	5.0% (4.6–5.5)	1.77 *** (1.37–2.29)
Low-income						
Guinea-Bissau	2.5% (0.6–9.9)	3.8% (3.0–4.9)	0.58 (0.14–2.46)	0.0% (0.0–6.6)	2.0% (1.4–2.8)	n/a
The Gambia	4.4% (2.1–8.9)	1.1% (0.7–1.6)	3.52 ** (1.49–8.28)	1.4% (0.4–4.8)	0.6% (0.3–1.0)	2.2 (0.58–8.37)
Chad	11.9% (8.4–16.7)	9.2% (8.0–10.4)	1.27 ()0.89–1.80)	6.1% (3.7–9.9)	4.2% (3.5–5.1)	1.41 (0.84–2.37
Togo	5.5% (3.1–9.5)	5.4% (4.2–7.0)	0.97 (0.52–1.81)	3.9% (1.9–7.6)	2.7% (1.8–4.0)	1.38 (0.62–3.07
Madagascar	15.3% (12.1–19.1)	11.4% (10.3–12.5)	1.38 ** (1.08–1.75)	6.8% (4.9–9.4)	4.3% (3.7–5.0)	1.50 * (1.06–2.13)
DR Congo	14.1% (8.8–21.8)	6.2% (5.2–7.3)	2.19 ** (1.34–3.59)	7.2% (3.4–14.5)	2.8% (2.1–3.6)	2.45 * (1.06–5.69)
Sierra Leone	14.7% (10.4–20.3)	13.7% (12.9–14.6)	0.99 (0.69–1.44)	10.7% (7.1–15.8)	10.0% (9.3–10.8)	0.96 (0.62–1.48)
Central African Republic	19.8% (15.9–24.3)	17.8% (16.0–19.8)	1.11 (0.87–1.41)	10.0% (7.5–13.4)	8.8% (7.6–10.2)	1.13 (0.81–1.57)
Malawi	9.7% (7.3–12.9)	7.9% (6.9–8.9)	1.23 (0.91–1.67)	4.4% (2.7–7.0)	3.3% (2.7–4.0)	1.31 (0.79–2.18)

Note: * *p* < 0.05, ** *p* < 0.01, *** *p* < 0.001; APPR = Prevalence rate ratio adjusted for age.

## Data Availability

All data used in this study are potentially available on request to UNICEF.

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
