# Peer review of "Disability among Women and Men Who Married in Childhood: Evidence from Cross-Sectional Nationally Representative Surveys Undertaken in 37 Low- and Middle-Income Countries"

_ijerph, 2022, doi:10.3390/ijerph20010088_

Round 1
Reviewer 1 Report
The paper undertakes an important topic concerning child marriage and disability. Associations between those have not been thoroughly explored yet by other researchers, so the paper introduces an interesting novelty in this field. The Authors base their analysis on the results from UNICEF’s Multiple Indicator Cluster Surveys what assures good data quality as the surveys are carried out on big and representative samples. The Authors focus on estimating the association between disability prevalence and child marriage in middle- and low-income countries according to the World Bank 2018 country classification based on per capita Gross National Income. Using various statistical methods (descriptive statistics, mixed effects multilevel multivariate modeling, regression analysis, meta analysis) the Authors evaluate the associations between the phenomena. The results give information that such associations are present and and this is the main cognitive value of the manuscript. However, there is no explanation of causality what gives the main limitation of the study as mentioned by the Authors. It would be interesting in the future work not only to identify the relationships but to explain them in-depth but it may be very difficult to achieve. Another crucial problem is that it is not known when disability occurred – in childhood or later, so the relation with child marriage cannot be studied in detail. Hence, the findings may become the starting point for further research.
I have some additional remarks:
1. There is a very big discrepancy in disability prevalence, i.e. from 4.9% (Turkmenistan) to 30.2% (Central African Republic). I am aware that there may be inequalities in this respect, but the difference is striking. It's hard to judge, but maybe the respondents interpreted the questions in the questionnaires differently? With cross-country studies there are often problems with translation and understanding. Do the Authors have any idea or knowledge how explain such a difference? If so it would be recommended to give a comment on it in the text.
2. In line 150 and 160 you use expression “Non-parametric correlation” without specifying which measure you use. Please give detailed information on it.
3. Line 191: is “marroiage” correct or is it misspelling?
Author Response
I have some additional remarks:
- There is a very big discrepancy in disability prevalence, i.e. from 4.9% (Turkmenistan) to 30.2% (Central African Republic). I am aware that there may be inequalities in this respect, but the difference is striking. It's hard to judge, but maybe the respondents interpreted the questions in the questionnairesdifferently? With cross-country studies there are often problems with translation and understanding. Do the Authors have any idea or knowledge how explain such a difference? If so, it would be recommended to give a comment on it in the text. We do not know the reasons behind the variations in the prevalence of disability. We believe that it is likely that many factors are involved including, as suggested, slightly differing meanings of translated items in different languages. Also likely to be involved are cultural and national differences in stigma associated with disability that may impact a person’s willingness to disclose such information, cultural and national differences in the incidence of disability and the survival of children and adults with disabilities. This is an interesting area, but not one that was central to our paper. As such, we believe including such speculations in the Discussion section of the present paper would not be appropriate.
- In line 150 and 160 you use expression “Non-parametric correlation” without specifying which measure you use. Please give detailed information on it. We have included this information.
- Line 191: is “marroiage” correct or is it misspelling? We have corrected this misspelling.
Reviewer 2 Report
Dear authors,
I find the topic of your research of extreme value. I have some comments that may help to improve how you present your results:
- In the introduction, it would be worth mentioning what is the international regulation about child marriage. For me, it is very interesting to see that the prevalence is a lot higher in countries in Latin America than in others where child marriage seems to be more of an issue. Perhaps a description of the law may give us an idea of what could be happening.
- In the methods section, I am a bit lost. You analyze large data sets (microdata) and estimate the prevalence of child marriage, and later the correlation of child marriage with disability. You undertake several statistical methods, but I do not know exactly why you do so. For example, you estimate multilevel models with Poisson regression.; however, Poisson is mainly used for count data where the distribution of the variable shows that the mean and variance are quite close. From the text, if I understand correctly, you correlate disability (levels) with child marriage (binary). Thus, I am not sure why you use such methods. In fact, if your goal is to estimate the correlation between these two indicators, given that you have a large sample size, simple methods give enough information.
My suggestion is to give a clear structure to the methods and use one statistical method. If you see the results have a huge variation, then provide an alternative method as a sensitivity analysis. Do not use several methods to test the results.
Also, within the methods, describe the variables of interest, and how are these measured. In this way, it is possible to follow up on the results.
Please, format the tables in such a way that is possible to read them without going into the text. Provide enough information in the notes of the tables.
- Please, report the results using the same structure as the methods section. In this way is possible to understand how are your results obtained.
Author Response
Dear authors,
I find the topic of your research of extreme value. I have some comments that may help to improve how you present your results:
- In the introduction, it would be worth mentioning what is the international regulation about child marriage. For me, it is very interesting to see that the prevalence is a lot higher in countries in Latin America than in others where child marriage seems to be more of an issue. Perhaps a description of the law may give us an idea of what could be happening. We have revised the introductory paragraph to clarify that: (1) then definition of child marriage as being marriage under 18 is a UN definition; and (2) notwithstanding this, child marriage is legally permitted in a significant percentage of the World’s countries.
- In the methods section, I am a bit lost. You analyze large data sets (microdata) and estimate the prevalence of child marriage, and later the correlation of child marriage with disability. You undertake several statistical methods, but I do not know exactly why you do so. For example, you estimate multilevel models with Poisson regression.; however, Poisson is mainly used for count data where the distribution of the variable shows that the mean and variance are quite close. From the text, if I understand correctly, you correlate disability (levels) with child marriage (binary). Thus, I am not sure why you use such methods. We use Poisson regression to generate unbiased estimates of relative risk. We have clarified this in the Approach to Analysis section and included a reference to support using Poisson regression in this manner. In fact, if your goal is to estimate the correlation between these two indicators, given that you have a large sample size, simple methods give enough information. We do not agree that the use of ‘simple methods’ in this context would be appropriate given the clustering of observations within countries. We note that the statistical reviewer (#3) did not raise any concerns about our approach to these statistical analyses.
My suggestion is to give a clear structure to the methods and use one statistical method. If you see the results have a huge variation, then provide an alternative method as a sensitivity analysis. Do not use several methods to test the results. We have revised the results so that our primary method is meta-analysis but given the high heterogeneity of many of the results we include multilevel modeling as a sensitivity analysis.
Also, within the methods, describe the variables of interest, and how are these measured. In this way, it is possible to follow up on the results. This in done in lines 72-112.
Please, format the tables in such a way that is possible to read them without going into the text. Provide enough information in the notes of the tables. We have revised the Tables to ensure that all acronyms and symbols are included in the Table Notes.
- Please, report the results using the same structure as the methods section. In this way is possible to understand how are your results obtained. We have revised the Approach to Analysis section to ensure it is consistent with the presentation of results.
Reviewer 3 Report
I was asked to review the statistical aspects of your submitted manuscript. I found the paper interesting and well written. I hope the conclusions are not lost in all the statistical discussion. Some comments follow:
1. Line 118 - Stata 16 should be referenced.
2. line 161 - Suggest 'moderate' rather than 'strong'. Correlations of .7 or greater are considered important because more than 50% of the variance is explained.
3. Lines 177-181 - Are these individuals that now have a disability? Do we know if they had a disability when they married?
4. Lines 283-294 - Good conclusion section. Suggest - should include a study to determine if children of childhood marriages are likely to have a disability.
Author Response
- Line 118 - Stata 16 should be referenced. We have included a reference for Stata 16.
- line 161 - Suggest 'moderate' rather than 'strong'. Correlations of .7 or greater are considered important because more than 50% of the variance is explained. We have made the suggested change.
- Lines 177-181 - Are these individuals that now have a disability? Do we know if they had a disability when they married? No, we do not know whether they had a disability when they married. This issue is discussed at length in paragraph commencing line 249.
- Lines 283-294 - Good conclusion section. Suggest - should include a study to determine if children of childhood marriages are likely to have a disability. We have made the suggested change.